# Generating Explanations to Understand Fatigue in Runners using Time Series Data from Wearable Sensors

**Bahavathy Kathirgamanathan** [1]  **Thu Trang Nguyen** [1]  **Brian Caulfield** [2]  **Georgiana Ifrim** [1]
**Pádraig Cunningham** [1]

## Abstract

Running while fatigued poses an increased risk of injury. Wearable sensors can be used to capture the running kinematics or running pattern as time series signals. The changes that happen in the running pattern due to fatigue, although prominent enough to increase the risk of injury, are generally only seen as subtle differences in the signal itself and hence are difficult to differentiate using purely visual inspection. In this paper, we introduce a time series dataset of motion capture data from runners before and after a fatiguing intervention. The total dataset consists of more than 5500 instances and was collected from 19 participants. The evaluation presented in this paper first looks at the effectiveness of a data aggregation technique called time series barycenters which is shown to improve classification performance. We evaluate and compare a set of classifiers and explanation methods for this problem, and select the most informative classifier and explanation for this dataset. We then present feedback from a domain expert on the insights offered by the explanations.

## 1. Introduction

The onset of fatigue in runners increases their risk of injury due to the higher impact accelerations and alterations in the overall running kinematics (Mizrahi, 2000). The presence of fatigue can cause a lack of control over joint motion and muscle contraction which causes this change in the running kinematics (Borgia et al., 2022). Furthermore, the changes

seen in runners are often specific to the individual as there is known to be no generalizable template of running technique and hence feedback should be individualised to each runner (Strohrmann et al., 2012).

Running kinematics for an individual can be captured using wearable sensors such as the Shimmer[1] sensors. The data captured from these sensors are time series in nature. The majority of research in this exercise classification domain uses extracted features from these time series as input into the models(O'Reilly et al., 2017). This requires a level of understanding regarding which features may be more important for a given classification task. Many problems in the exercise domain such as the task considered in this paper, where we predict fatigue in runners, have subtle differences between classes which may not be noticeable by the eye alone. Using time series classification techniques allows the signal to be used in a time series format at a single stride level without needing feature extraction. Additionally, this allows interpretable classification techniques to be used to highlight the particular regions of interest within a time series. This information is valuable as it can inform the runner when they are changing their form and hence can help prevent fatigue related injuries.

The data captured from the sensors can be noisy and due to the individualised nature of the running pattern, automated segmentation procedures are not always completely accurate. This can impact the classification performance and the stability of the interpretations made. Hence we employ a barycenter averaging strategy (Cuturi & Blondel, 2017) which aggregates and smoothes multiple strides together which we expect will improve the classification performance and improve the overall stability of the explanations.

The key contributions of this paper are as follows:

- We present the methods used to capture, segment, and classify motion capture data for a running fatigue prediction task. Furthermore, we release this time series dataset for use by the machine learning community. To

[1] School of Computer Science, University College Dublin, Dublin, Ireland [2] School of Public Health, Physiotherapy and Sports Science, University College Dublin, Ireland. Correspondence to: Bahavathy Kathirgamanathan <bahavathy.kathirgamanathan@ucdconnect.ie>, Pádraig Cunningham <padraig.cunningham@ucd.ie>.

*Workshop on Interpretable ML in Healthcare at International Conference on Machine Learning (ICML)*, Honolulu, Hawaii, USA. 2023. Copyright 2023 by the author(s).

[1] http://shimmersensing.com

access the anonymised data, see `https://zenodo.org/record/7997851#.ZJ2XenbMI2x`.

- We evaluate the classification performance of this running fatigue prediction dataset using various models that are used for interpretable time series classification. Furthermore, we evaluate the impact on classifier performance when using a barycenter averaging technique as a smoothing strategy on the wearable sensor data.

- We produce explanations for the time series data and discuss the insights into the impact of fatigue in runners that can be deduced from these explanations.

## 2. Related Work

### 2.1. Impact of Fatigue on Runners

Running has become a popular form of exercise over the past few decades and this has led to an increase in the rate of running-related musculoskeletal injuries (Tonoli et al., 2010). There are many factors that can cause these injuries and often they are due to faults in the technique. Faults in a runners technique can start to happen or get exaggerated as they get fatigued as they start to lose control over their running form. This lack of control in the technique is known to be a primary contributor to running related injuries (Willems et al., 2006). Altering the kinematics while running is often an attempt by the runner to minimise the overall metabolic cost due to fatigue (Hunter & Smith, 2007). Hence it is important to identify when a runner's form is beginning to change. Being able to objectively measure these alterations is difficult and often runners are required to rely on their own assessment of running form (Buckley et al., 2017). Wearable sensors capture these kinematic changes hence allowing an objective analysis of running form to be made.

### 2.2. Time Series Classification

A time series is a time-based sequence of observations, $x_i(t); [i = 1, \ldots, m; t = 1, \ldots, p]$, where $x_i(t)$ is the observation for the $i^{th}$ dimension at time point $t$. The time series is univariate when $m = 1$ and multivariate when $m \geq 2$.

Time Series Classification (TSC) techniques are classed as distance based, shapelet based, symbol based, deep learning methods and ensembles. Distance based techniques are known to perform well and the common benchmark of 1-Nearest Neighbour (1-NN) with dynamic time warping (DTW) is one of the most popular TSC approaches (Bagnall et al., 2017). 1-NN DTW however has a high computational complexity and sometimes faces challenges to the accuracy in the presence of noise (Schäfer, 2016). Time series shapelets are subsequences of the data which capture the portion of the time series maximally representative of the

class. There have been various implementations of shapelets such as the original work by Ye and Keogh (Ye & Keogh, 2009), Shapelet Transform(Lines et al., 2012) and Learning Shapelets(Grabocka et al., 2014). Shapelets, however are not widely used as they can be computationally expensive (Ruiz et al., 2020).

Symbol based techniques include Mr-SEQL (Le Nguyen et al., 2019), WEASEL+MUSE (Schäfer & Leser, 2017) and BOSS (Schäfer, 2015). Symbol based techniques work by passing a sliding window over each time series and representing each window with a word of symbols. This transform is done using techniques such as Symbolic Aggregate Approximation (SAX)(Lin et al., 2007) or Symbolic Fourier Transform (SFA)(Schäfer & Högqvist, 2012). Although these methods are known to work well, they often suffer a long runtime. Most recent ensemble methods work by transforming the time series into a new feature space. From this idea, COTE (Collective of Transformation based ensembles) was developed which works by ensembling different classifiers over different time series representations (Bagnall et al., 2015) This technique was later extended as HIVE-COTE (Hierachical Vote system) which uses a hierachical structure based on probabilistic voting (Lines et al., 2018). HIVE-COTE has been shown to achieve high accuracy, however has a large computational complexity.

Deep learning techniques such as ResNet and Fully Convolutional Networks (FCN) have also been shown to perform well for time series classification. However, the most popular and current state-of-the art technique is ROCKET (Dempster et al., 2020), which is a method that borrows ideas from deep neural networks where a simple linear classifier is trained on random convolutional kernels. This method has been shown to achieve state-of-the-art performance whilst also maintaining a lower computational load than other state-of-the-art techniques.

For the evaluations in this paper we have selected four techniques, Ridge regression, Rocket, 1-NN-DTW, and Mr-SEQL as detailed in Section 4.

### 2.3. Time Series Barycenter

A time series barycenter is an average measure of a collection of time series. Typically time series averaging strategies are classified as local or global. Local strategies use pairwise averaging where a collection of series are iteratively averaged into a single average series. Local averaging is dependent on the order in which samples are aggregated and hence changing the order can give a different result. Recent advances look at global average strategies which compute the average of the set of time series simultaneously. These averaging strategies generally use a similarity metric to find the distance between the series. Dynamic Time Warping (DTW)(Sakoe & Chiba,

1978) is one of the most common similarity metrics used for time series and hence is used in most time series barycenter calculation techniques. DTW maps the time series in a non-linear way and works to find the optimal alignment between the two series. Two popular global averaging strategies are the DTW Barycenter Averaging (DBA) proposed by Petitjean *et al.* (Petitjean et al., 2011) and the Soft-DTW Barycenter proposed by Cuturi *et al.* (Cuturi & Blondel, 2017).

**DTW Barycenter Averaging (DBA)** computes the optimal average sequence within a group of series in DTW space by minimising the sum of the squared DTW distance between the average sequence and the group of series (Petitjean et al., 2011). To compute the DTW barycenter of a set of time series $D$ is the optimisation problem outlined in Equation 1.

$$\min_{\mu} \sum_{x \in D} DTW(\mu, x)^2 \qquad (1)$$

Where $x$ is a series belonging to the set $D$ and $\mu$ is a candidate barycenter. The DTW distance between each time series and a temporary average sequence (candidate barycenter) is calculated and the temporary average sequence is updated until the optimisation criteria is met(Shi et al., 2019).

**Soft-DTW Barycenter** computes the average sequence within a group of series by minimising the weighted sum of the Soft-DTW distance between the average sequence and the group of series. Soft-DTW is an extention of the DBA method where the min operator is replaced by the soft-min. This has the advantage of being differentiable with respect to all of its inputs. Soft-DTW also has the advantage where it considers all possible alignments. Soft-min can be computed as shown by Equation 2 (Tavenard et al., 2017).

$$\text{softmin}_{\gamma}(a_1, ..., a_n) = -\gamma \log \sum_{i} e^{-a_i/\gamma} \qquad (2)$$

$\gamma$ controls the smoothing and hence as $\gamma \to 0$, the result gets closer to that of DTW.

Previous literature has shown Soft-DTW barycenter averaging to be an effective way to aggregate time series data as it preserves the key features well (Kathirgamanathan et al., 2022b) and hence this technique will be employed in our evaluations.

## 2.4. Time Series Interpretation

Time Series data often comes from domains such as healthcare where having explainable models is of importance. Many time series interpretation methods are based on explaining through visualisation and feature importance rather than instance based explanations(Kenny et al., 2021). Instance based techniques such as 1-NN DTW although often used as a benchmark for time series classification may not be ideal for explanation as the nearest labelled neighbour can be found, however, the specific parts of the time series that are more influential for a given classification task are unknown(Le Nguyen et al., 2019). Shapelet based techniques are by nature interpretable as they pick up on the section of the time series that is maximally representative of a class; however, they are computationally inefficient. Deep learning methods, although black box by nature, have methods which allow explanations to be made based on the class weights. A popular method for interpretation of time series is the use of Class Activation Maps (CAM)(Zhou et al., 2016) which is a saliency method that highlights discriminating parts of the time series.

Many model-agnostic methods used outside the field of time series generate local explanations use a perturbation strategy where features are slightly altered to gain insight into which features are more relevant to a model. For this, mapping functions need to be defined to direct how the perturbations should be done. LIME is one of the first techniques which uses this perturbation strategy (Ribeiro et al., 2016) which works for image, text, and tabular data. SHapeley Additive exPlanations (SHAP) (Lundberg & Lee, 2017) is a current widely used technique which is a unified measure of feature importance. TimeXplain is a recent development which works with SHAP by defining mappings that work on the time and frequency domain (Mujkanovic et al., 2020). Hence TimeXplain makes SHAP usable with time series data.

## 3. Data Collection and Processing

### 3.1. Experimental Setup and Protocol

Nineteen recreational runners were recruited to participate in this study. The participants were all free of lower limb injury and were regular runners (at least 2 runs/week). The study protocol was reviewed and approved by the human research ethics committee at University College Dublin.

A single lumbar mounted Shimmer Inertial Measurement Unit (IMU) was mounted on each participant while they completed the task in three segments. The tests were done on an outdoor track typically used for running. In the first segment, the participant completed a 400m run at a comfortable pace. In the second segment, the participant completed a beep test(Léger et al., 1988), which is a multi-stage fitness test where the runner is required to run continuously between two points 20m apart following an audio which produces 'beeps' to indicate when the person should start running from one end. As the test progresses, the interval between the 'beeps' reduces and hence the runner will be

required to increase their pace. The test ends when the runner is unable to continue or when they are unable to keep up with the pace of the 'beeps'. For this study, this was used as the fatigue intervention. In the final stage, the participant was required to repeat the 400m run, this time in their fatigued state. The fatigued run was completed immediately after the completion of the beep test.

The IMU sensor captured acceleration, angular velocity and magnetometer data. The data was collected as three long time series (non-fatigued run, beep test, and fatigued run) and was then segmented into individual strides as described in Section 3.2. The sampling rate of the sensors was set to 256Hz. Altogether from the sensors, there were nine signals extracted: Acceleration (X,Y,Z), Angular Velocity (X,Y,Z), and Magnetometer (X,Y,Z). From the acceleration signals, the magnitude acceleration was derived ($A_{mag} = \sqrt{X^2 + Y^2 + Z^2}$). This signal was primarily used for the analysis presented in this paper. Our analysis focuses on the two 400m runs which we frame as a binary classification task to distinguish between Non-Fatigued and Fatigued.

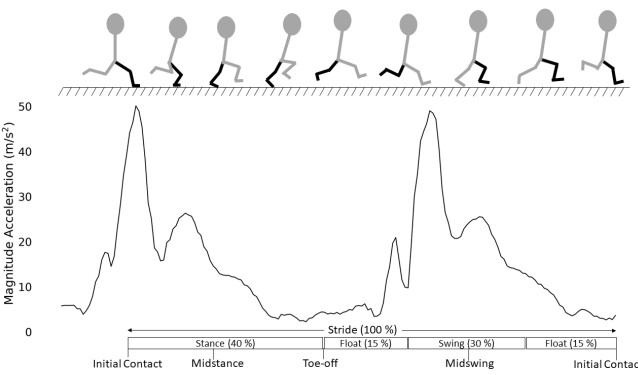

*Figure 1.* Breakdown of a single stride

### 3.2. Segmentation and Pre-Processing strategy

The data was segmented into individual strides using a segmentation protocol as follows. Each instance was a stride or two steps (See Figure 1). To segment the data into individual strides, the acceleration in X-direction was first used to identify the left foot so each stride begins with the left limb. The peak point was identified and the minimum before the peak was used as the point for segmentation. The breakdown of a single stride is shown in Figure 1 where the first peak roughly corresponds to the left foot initial contact point. The segmented strides were then resampled to the length of the median stride as the signals are required to be of equal length for some of the interpretation strategies used later in this paper. The segmentation and processing were done using Python semi-automated scripts.

Figure 2 summarises the procedure that was used for the data

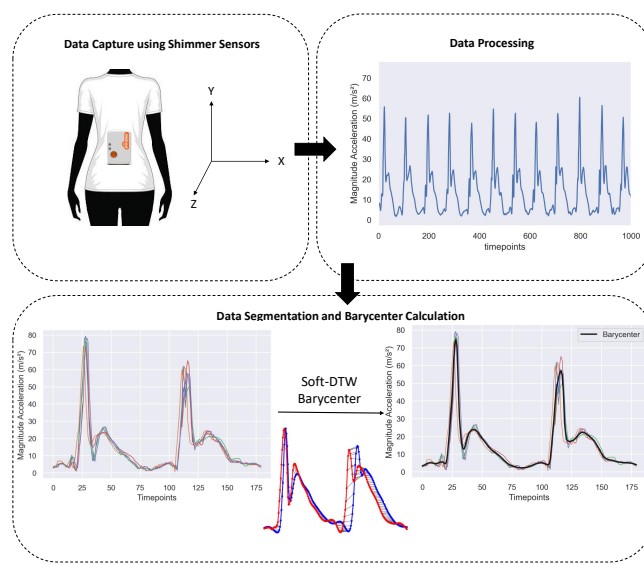

*Figure 2.* Data collection and processing protocol; Data was initially collected using Shimmer sensors, The signals were segmented into individual strides and aggregated 5 strides at a time using a rolling Soft-DTW barycenter averaging technique.

collection and processing. Following data collection and segmentation, we use the barycenter aggregation technique as a smoothing technique for the running data. We employ a rolling barycenter calculation over every five strides. An example of the five strides and its corresponding barycenter can be observed in Figure 2. To check the effectiveness of using barycenters to represent the data, we evaluate the classification performance of the barycenters against the full set of strides in the next section (Section 4).

## 4. Classification Performance

The classification was performed at an individual participant level as previous research suggests global classifiers do not work well in this domain due to differences in the running style between individuals (Kathirgamanathan et al., 2022a). Hence, we have 19 separate datasets with approximately 290 strides or instances per dataset. The task is posed as a binary classification task where we predict fatigued strides against non-fatigued strides. The four time-series classification models considered in the evaluation were selected for accuracy and interpretability:

- **Ridge regression (RidgeCV)**[2] is a linear classifier and is used as one of the simplest models and takes the time series data as tabular data. Ridge regression can work well on time series data where the values rather than the shape of the data are of influence.

[2]https://scikit-learn.org/stable/modules/generated/sklearn.linear_model.RidgeCV.html

- **Rocket** is a current state-of-the-art technique for time series classification (Dempster et al., 2020). Rocket works by generating random convolutional kernels which are then convolved along the time series to produce a feature map. These features are then input into a simple linear classifier such as a Ridge Classifier or Logistic Regression. In our evaluation, the default 10,000 kernels and random state of 0 are used.

- **1-Nearest Neighbour (1-NN) with Dynamic Time Warping (DTW)** has been used commonly as a benchmark classifier for time series. It is known to be one of the more reliable and simple approaches for time series classification (Bagnall et al., 2017).

- **Mr-SEQL**(Le Nguyen et al., 2019) is a linear classifier which uses symbolic features extracted from the time series. Mr-SEQL is an interpretable model and has been used for explainable AI in Time Series research. Hence it has been included in this evaluation.

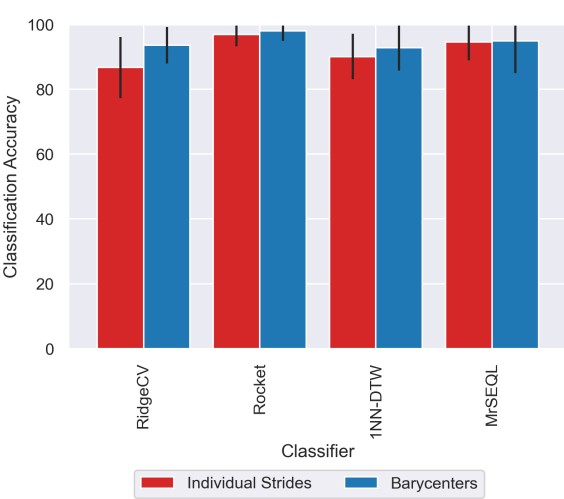

*Figure 3.* Average accuracy using individual strides versus barycenters across the 19 subjects for the four selected models

Each dataset was split with the first two-thirds of each 400m run being used as the training data and the last one-third used as test data which was done to preserve the time series nature of the problem. On average across the participants there were 192 training instances and 98 test instances per participant. The classification models were run on both the individual strides as well as the aggregated barycenter signals. The last four barycenters in the train data were removed to prevent any training data being leaked into the test data.

A summary of the results are shown in Figure 3 and the full set of results is presented in the git repository with

the anonymised data [3]. Overall Rocket performed the best but all classifiers were able to produce an average accuracy across the subjects above 0.8. Using the barycenters instead of the individual strides slightly improved performance suggesting that the barycenters could be a good smoothing technique to use.

## 5. Explanation

To provide insights into the impact of fatigue on the runners, we present personalised explanations for selected participants. We aim to investigate if interpretable time series techniques can successfully identify the more discriminative regions of the time series and if these regions can be explained in a real-world context.

We select four participants and present a personalised explanation for each participant. We assess a suitable explanation technique using a ranking method as detailed in Section 5.1. The participants were selected on the basis of covering a range of scenarios of how fatigue impacts runners. We select three participants who show increased peak accelerations after fatigue which is a commonly seen trend amongst many participants and one participant who does not show this increase.

### 5.1. Strategy for selecting the explanation method

#### 5.1.1. EXPLANATION METHODS.

As we are interested in exploring the difference among fatigued and non-fatigued runners using the raw time series signals, we select local time-based explanation methods that specifically show the critical moments (in time) that differentiate the two classes. This explanation is often presented in the form of a saliency map, highlighting the discriminative areas of the data. It is usually represented as a set of importance weights (from 0 to 100), one weight for each point in the target time series. In this explanation assessment, we evaluate 8 popular methods (Table 1). Of the selected methods, Gradient SHAP (Lundberg et al., 2018) and Integrated Gradient (Sundararajan et al., 2017) are commonly used in explaining deep models for image classification, using the gradients of the trained models. SHAP (Lundberg & Lee, 2017) and LIME (Ribeiro et al., 2016) are model-agnostic, post-hoc explanation methods that can be used to explain any classifiers. In our evaluation, MrSEQL-LIME, MrSEQL-SHAP, ROCKET-LIME, and ROCKET-SHAP are results of LIME and SHAP explanations for the MrSEQL(Le Nguyen et al., 2019) and ROCKET (Dempster et al., 2020) classifiers. Intrinsic weights from MrSEQL (Le Nguyen et al., 2019) (time-series specific classifier) and Ridge Regression classifier (general classifier) are also

---

[3]https://zenodo.org/record/7997851#.ZJ2XenbMI2x

| Explanation Method | Rank for Participant | | | | Overall Rank | |
|---|---|---|---|---|---|---|
| | **07** | **08** | **17** | **18** | Avg | Number (1-9) |
| Gradient SHAP | 6 | 3 | 6 | 7 | 5.50 | 5 |
| Integrated Gradient | 3 | 1 | 9 | 8 | 5.25 | 4 |
| MrSEQL-LIME | 5 | 5 | 5 | 3 | 4.50 | 3 |
| ROCKET-LIME | 9 | 6 | 3 | 4 | 5.50 | 5 |
| **MrSEQL-SHAP** | **1** | **2** | **1** | **1** | **1.25** | **1** |
| ROCKET-SHAP | 2 | 4 | 8 | 2 | 4.00 | 2 |
| MrSEQL-SM | 4 | 7 | 2 | 5 | 4.50 | 3 |
| RidgeCV-SM | 8 | 9 | 7 | 9 | 8.25 | 7 |
| Random | 7 | 8 | 4 | 6 | 6.25 | 6 |

*Table 1.* Explanation evaluation ranking results. MrSEQL-SHAP has the best rank and is thus the most informative explanation method for this problem. The rank of each explanation method is computed for each participant dataset, and then averaged across participants.

added to the explanation methods to be evaluated. A random explanation series (randomly generated importance weights) is used as a sanity check to filter out any ineffective explanation methods.

### 5.1.2. SELECTION STRATEGY.

We apply the explanation selection strategy presented in (Nguyen et al., 2023) to evaluate the selected candidate explanation methods. This strategy aims to estimate the impact of discriminative data areas (identified by an explanation method) by perturbing the data in that area and measuring how a time series classifier (referee classifier) responds to the perturbation. A good explanation method that correctly identifies such discriminative areas will theoretically trigger a more significant drop in accuracy of the referee classifier. This strategy uses a committee of referee classifiers and a combination of different perturbation strategies to reduce possible bias and enhance robustness in the evaluation.

In our assessment, the committee of Referee Classifiers includes ROCKET (Dempster et al., 2020), Mr-SEQL (Le Nguyen et al., 2019), 1NN-DTW (Bagnall et al., 2017) and Ridge Regression on datasets perturbed with mean and Gaussian profile from either entire dataset or specific time point (Mujkanovic et al., 2020).

### 5.1.3. RESULTS & SELECTED EXPLANATION

We present the results of our time series explanation evaluation in Table 1. We note that by evaluating and ranking 8 explanation methods on the 4 participants we analyse in depth (with ids: 07, 08, 17, 18), the most informative

explanation method for our data is MrSEQL-SHAP.

### 5.2. Discussion and Feedback from the Explanations

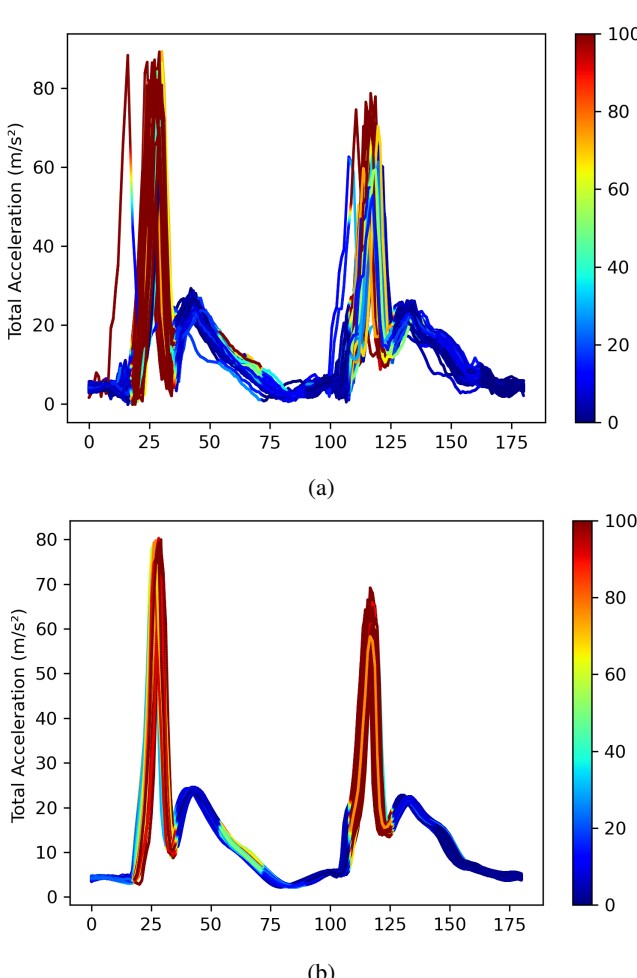

*Figure 4.* (a) Mr-SEQL-SHAP explanation on the individual strides (Fatigued Class), (b) Mr-SEQL-SHAP explanation on the rolling barycenters calculated from the individual strides (Fatigued class) for Participant 8.

The selected explanation technique, Mr-SEQL-SHAP was used to generate explanations for the four selected participants and these were discussed with our domain expert to gain insights into how fatigue affected these runners. The aggregated barycenter strides were used for the evaluations due to improved classification performance. Indeed the barycentres may also help to improve the clarity of the explanations. Figure 4 shows saliency maps showing a Mr-SEQL-SHAP explanation for both the individual strides and the aggregated barycenter strides for participant 8 and it can be seen that the barycenters aid in visualising the saliency and potentially improving the stability of the predictions. A reason for this could be that many of these interpretation techniques work well when the data is aligned well. Al-

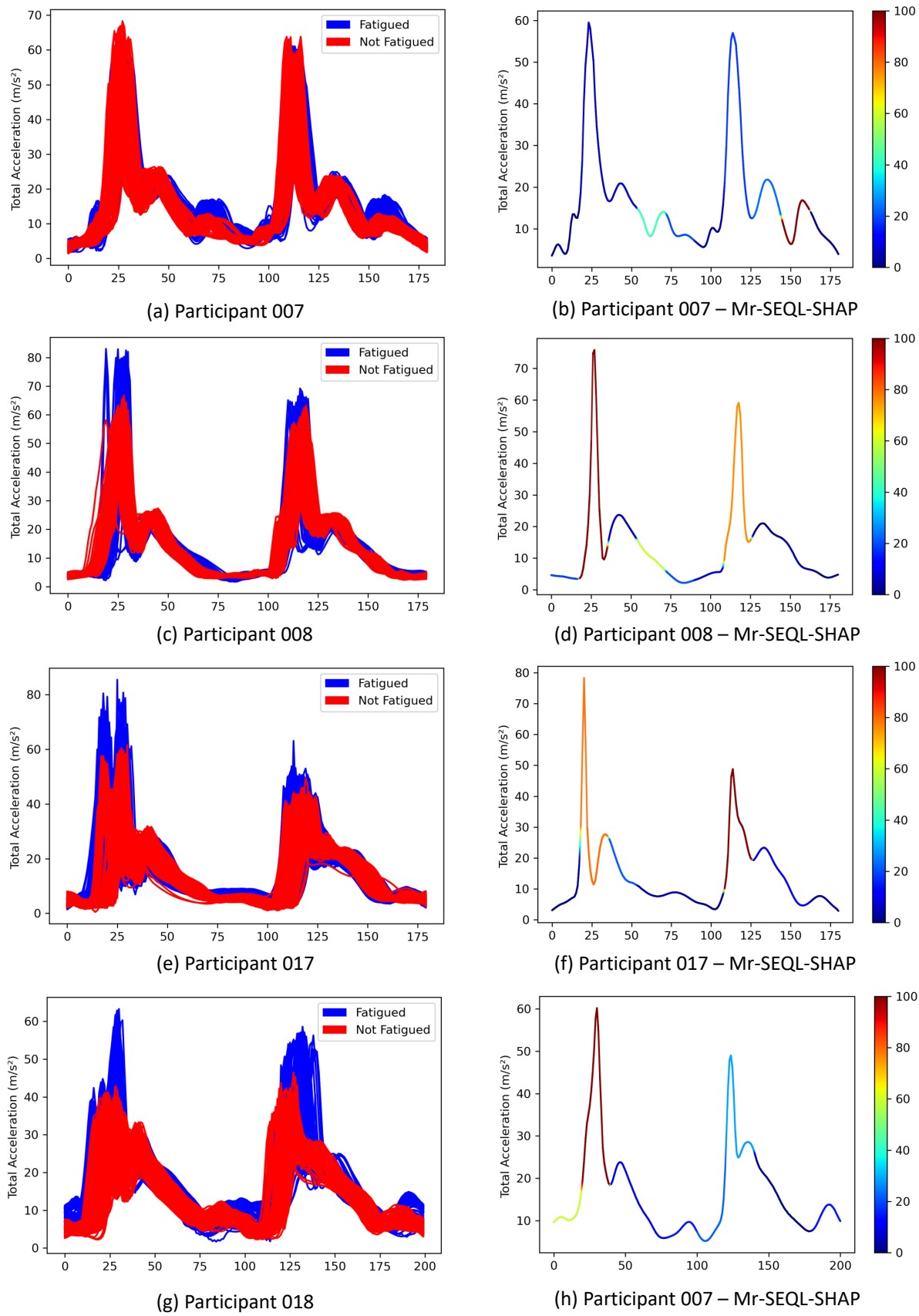

*Figure 5.* Left: A map of the fatigued versus not-fatigued instances for the selected four participants. Right: Mr-SEQL-SHAP generated saliency maps for a sample of a fatigued stride

though the process of segmentation makes the nature of the initial data reasonably well aligned, the barycenter further aids in this overall alignment of the time series.

Figure 5 on the left shows a mapping of all the fatigued and non-fatigued strides against each other and a saliency map of a sample fatigued stride produced from Mr-SEQL-SHAP for each of the four participants. The red highlighted regions in the saliency map suggest that region contributed more towards the classification decision.

The most common effect of fatigue was an increase in the peak acceleration point which corresponds to the initial contact where the foot strikes the ground. This suggests that most people lose control during the loading stage. This supports previous research which suggests that the natural damping of runners reduces with fatigue (Willems et al., 2006) which would affect the loading stage or the weight absorption stage of running which is observed around the peaks in our data.

The insights from the highlighted regions (Figure 5) of each participant can be summarised as follows:

**Participant 8 and 18:** This is the most straightforward case and perhaps the most common in the dataset. These participants have a fairly symmetric gait - the left and right peaks are similar. Perhaps there is a slightly higher peak acceleration on their left (non-dominant) leg. However, there is a clear asymmetry in the fatigued state and this is highlighted in dark red in the saliency maps. For participant 8 the saliency map highlights the second peak but to a lesser extent. So for both participants the feedback is that fatigue shows up particularly in the non-dominant leg resulting in loss of control during the loading stage.

**Participant 17:** This participant is similar to 8 and 18 except that there is already evidence of asymmetry in the non-fatigued strides. The asymmetry shows up as higher peak acceleration on the left leg. This asymmetry is accentuated when fatigued. Both peaks are highlighted but the saliency map gives more weight to the right leg in this case.

**Participant 7:** This is the most interesting participant in some respects. We can see in Figure 5a that there is no difference in peak impact between the fatigued and non-fatigued states. So there is no evidence of loss of control during the loading stage. Nevertheless the classifiers are able to distinguish between the two classes with very high accuracy. The saliency map in Figure 5b indicates that the discriminative region is later in the stride; it is during the unloading phase where the leg is preparing to go into flight. This is interesting as although the participant was able to maintain their impact acceleration which is known as an

indicator of fatigue, they are still modifying their overall running gait in a way that is less obvious to an observer.

Overall the discriminatory regions that were identified by the Mr-SEQL-SHAP technique supported the overlay plots in Figure 5 as the regions which were clearly different in the overlay plots were highlighted in the explanations. However, the saliency plots also provide more information as they were able to highlight the particular limb that was contributing more to the classification task. Furthermore, it should be noted that there were some minor discrepancies in the segmentation due to the automated nature and this can be seen in the bi-modal behaviour in participant 17 (Figure 5e) where there are two apparent points that were interchangeably selected during segmentation. Despite this, the models were easily able to classify accurately and select meaningful regions as discriminatory. This adaptability is important due to the personalised nature of running, where a one-fit-all solution is not feasible.

## 6. Conclusions and Future Work

In this paper, we present the protocol for collecting and processing a dataset of wearable sensor data from runners in their normal and fatigued states. We then evaluate the utility of using a barycenter averaging strategy as a smoothing and aggregation strategy to improve overall classification and explanation. Finally, we identify a suitable explanation technique and apply this to our data to gain insights into how fatigue impacts runners.

The use of barycenters to aggregate the time series proved to improve the classification performance as well as improve the visual aspect of the explanation. A variety of explanation techniques were also investigated and Mr-SEQL-SHAP came out to be the most effective way to explain this particular dataset. Combining the explanations generated from the Mr-SEQL-SHAP method with domain specific knowledge, some insights into how fatigue impacts runners were made. It was observed that many runners lost control during the loading stage, which is around the peak impact point. However this was not the case for every participant, and hence shows the importance of having personalised feedback systems. These feedback systems could be very helpful to runners in providing personalised feedback on the changes they are making to their overall running kinematics.

The results showed that Mr-SEQL with SHAP was able to highlight domain meaningful discriminative regions of the time series for the magnitude of acceleration data. However, the sensors have multivariate data, and looking at the other signals, may be useful in providing further insights into the data. Furthermore, there is limited research that looks at explanations for multivariate time series data and hence an

interesting avenue to investigate would be to extend this work to a multivariate case.

## Acknowledgements

This work has emanated from research conducted with the financial support of Science Foundation Ireland under the Grant numbers 18/CRT/6183. For the purpose of Open Access, the author has applied a CC BY public copyright license to any Author Accepted Manuscript version arising from this submission.

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
