# OpenReview forum: "Generating Explanations to understand Fatigue in Runners Using Time Series Data from Wearable Sensors"
_ICML.cc/2023/Workshop/IMLH — IMLH 2023 Poster_

### Official Review · Reviewer_Tbxp · 2023-06-11
**An interesting applicaiton and dataset**

**Rating:** 6
**Confidence:** 4

**Review:**

This long 8-page paper proposes a method for collecting and processing a dataset of wearable sensor data from runners in their normal and fatigued states. In this paper, the authors also use the barycenter averaging strategy to improve the classification and explanation. Finally, they identify the most convincing technique.

Strengths:

•	This paper is well-organized and the writing is clear. The illustrations in the paper are very intuitive, clear, and convincing.

•	The motivation is strong. The paper addresses an important issue in its research, contributing a new dataset.

•	The explanation of the experiments and results in the paper is comprehensive. There are many comparative experiments in the paper, and the experiments are conducted thoroughly.

Weaknesses/Suggestions:

•	The authors are encouraged to compare more up-to-date explanation methods [1,2,3] for a more informative comparison.

•	It will also be interesting to investigate the aggregation of different explanation methods.


[1] Sivill, Torty and Peter A. Flach. “LIMESegment: Meaningful, Realistic Time Series Explanations.” International Conference on Artificial Intelligence and Statistics (2022).
[2] Li, Peiyu et al. “Motif-guided Time Series Counterfactual Explanations.” ArXiv abs/2211.04411 (2022): n. pag.
[3] Boubrahimi, Soukaina Filali and Shah Muhammad Hamdi. “On the Mining of Time Series Data Counterfactual Explanations using Barycenters.” Proceedings of the 31st ACM International Conference on Information & Knowledge Management (2022): n. pag.

---

### Official Review · Reviewer_qCPi · 2023-06-13
**solid work on classifying and interpreting fatigue in runners**

**Rating:** 6
**Confidence:** 4

**Review:**

This paper collects a time-series dataset of motion captured by wearable sensor from runners, evaluates various models for classifying and interpreting fatigue vs non-fatigue strides and discusses possible explanations for fatigue based on the interpretation results.

While the technical novelty is limited, the protocol and pipeline are complete and comprehensive, and the results are insightful. I vote for acceptance for this paper.

---

### Official Review · Reviewer_5GsY · 2023-06-15
**This paper investigates changes in running patterns in fatigued and non-fatigued states, with the purpose of predicting and explaining these patterns using time series prediction.**

**Rating:** 6
**Confidence:** 5

**Review:**

The paper is well-written and clear, and also trained and tested in a well-defined dataset. However, while this paper identifies meaningful differences in running patterns during fatigue for single-subject data, it is still not clear how these explanations can be meaningfully aggregated to say something specific about the relationship between fatigue and injury. This limits the significance of this work.

Pros:

•	The data is representative for testing fatique, given the longitudinal design with an intervention between the two states (fatigue and non-fatigue)

•	The paper clearly explains the methods used to capture, segment and classify the motion data to predict fatigue stats.

Cons:

•	Table 1: this reviewer would like to see the results from all participants, to verify that the ranking is generalizable across participants.

•	Despite the authors trying to avoid leakage between the training and test set by removing samples, this cannot be avoided when training and testing on the same participant. This will consequently overestimate the obtained accuracy. The authors should change this cross-validation strategy to instead train on certain subjects and predict on other subjects, to avoid violation of independence.

•	According to Figure 3, there seems to be no statistical differences between individual strides and barycenter, thus limiting the papers conclusion on page 8 “The use of barycenters to aggregate the time series proved to improve the classification performance”. This claim is not substantiated by any statistical test.

---

### Meta-Review · Area_Chair_psAb · 2023-06-20

**Recommendation:** Accept (Poster)
**Confidence:** 5

**Metareview:**

The paper presents a well-written study on classifying and interpreting fatigue in runners based on motion data collected from wearable sensors. Reviewers acknowledge the completeness and comprehensiveness of the protocol and pipeline, the paper's clear organization, strong motivation, and insightful results. Nevertheless, comparing more up-to-date explanation methods and investigating the aggregation of different techniques is strongly recommended. A clear accept from all reviewers.

---

### Decision · Program_Chairs · 2023-06-20

Accept (Poster)